# Gender and Race-Based Health Disparities in COVID-19 Outcomes among Hospitalized Patients in the United States: A Retrospective Analysis of a National Sample

**DOI:** 10.3390/vaccines10122036

**Published:** 2022-11-29

**Authors:** Suman Pal, Karthik Gangu, Ishan Garg, Hina Shuja, Aniesh Bobba, Prabal Chourasia, Rahul Shekhar, Abu Baker Sheikh

**Affiliations:** 1Department of Internal Medicine, University of New Mexico Health Sciences Center, Albuquerque, NM 87106, USA; 2Department of Internal Medicine, University of Kansas Medical Center, Kansas City, KS 66103, USA; 3Department of Medicine, Karachi Medical and Dental College, Karachi 74700, Pakistan; 4Department of Medicine, John H Stronger Hospital, Cook County, Chicago, IL 60612, USA; 5Department of Medicine, Mary Washington Hospital, Frederickburg, VA 22401, USA

**Keywords:** vaccine inequity, gender disparity, COVID-19, race disparity, NIS data

## Abstract

COVID-19 has brought the disparities in health outcomes for patients to the forefront. Racial and gender identity are associated with prevalent healthcare disparities. In this study, we examine the health disparities in COVID-19 hospitalization outcome from the intersectional lens of racial and gender identity. The Agency for Healthcare Research and Quality (AHRQ) 2020 NIS dataset for hospitalizations from 1 January 2020 to 31 December 2020 was analyzed for primary outcome of in-patient mortality and secondary outcomes of intubation, acute kidney injury (AKI), AKI requiring hemodialysis (HD), cardiac arrest, stroke, and vasopressor use. A multivariate regression model was used to identify associations. A *p* value of <0.05 was considered significant. Men had higher rates of adverse outcomes. Native American men had the highest risk of in-hospital mortality (aOR 2.0, CI 1.7–2.4) and intubation (aOR 1.8, CI 1.5–2.1), Black men had highest risk of AKI (aOR 2.0, CI 1.9–2.0). Stroke risk was highest in Asian/Pacific Islander women (aOR 1.5, *p =* 0.001). We note that the intersection of gender and racial identities has a significant impact on outcomes of patients hospitalized for COVID-19 in the United States with Black, Indigenous, and people of color (BIPOC) men have higher risks of adverse outcomes.

## 1. Introduction

As of October 2022, COVID-19 has resulted in more than 97 million cases and >1 million deaths in United States [1]. Almost 21,000 patients still remain hospitalized with COVID-19 nationally [1]. Disparities in health outcomes among racial identities have been identified in the United States [2,3,4]. These are differently affected by various other demographic factors, including gender [2,5]. The burden of COVID-19 has the potential to exacerbate existing inequities in healthcare outcomes. Early studies on COVID-19 noted racial and gender disparities in outcomes [5,6]. However, these studies were limited by a smaller sample size, which limited analysis of groups which represent a minority of the population. To bridge this gap, we undertake this study using a national data sample. Using data available in Nationwide Inpatient Sample (NIS), we aim to study the outcomes of COVID-19 hospitalization through the intersectional lens of gender and racial/ethnic identity.

## 2. Materials and Methods

### 2.1. Data Source

This retrospective study utilized the Agency for Healthcare Research and Quality (AHRQ) 2020 NIS dataset, which is based on hospitalizations from 1 January 2020 to 31 December 2020 [7]. All patients who were 18 years of age and older and were admitted to the hospital and had a diagnosis of COVID-19 were included in the study. Patients with missing information about race and or sex were excluded from the study. International classification of diseases 10th—clinical modification (ICD-10-CM) codes were used to retrieve patient samples with comorbid conditions, and ICD-10 procedure codes were used to identify inpatient procedures. A detailed code summary is provided in Appendix A. 

### 2.2. Covariates

NIS data sample contains data regarding in-hospital outcomes, procedures, and other discharge-related information. Variables were divided into patient level, hospital level, and illness severity.

a.Patient level: Age, comorbidities, insurance status, income in patient’s zip code, disposition.b.Hospital level: Location, teaching status, bed size, region.c.Illness severity: Length of stay (LOS), mortality, hospitalization cost, Elixhauser comorbidity score, in-hospital complications, mechanical ventilation, vasopressor use, acute kidney injury (AKI) requiring hemodialysis (HD).

### 2.3. Study Outcomes

The primary outcome assessed was racial and gender impact on in-hospital mortality. Secondary outcomes were (a) intubation rate, (b) AKI, (c) AKI requiring HD, (d) cardiac arrest, (e) vasopressor use, (f) stroke, and (g) length of stay.

### 2.4. Statistical Methods

STATA 17 (StataCorp LLC, College Station, TX, USA) was utilized for statistical analysis. 

NIS 2020 data were sampled from 49 statewide organizations covering 98% of the US population and 97% of discharges from US hospitals. Sampled data included approximates to 20% or ~7 million discharges collected for that year. To obtain national estimates and extrapolate results for particular disease, weights were provided, and they were calculated using the formula below. Ideally, each disease discharge weight should be 5, but the weight significantly varies due to random sampling, and hence we used weights provided using survey analysis to provide national estimates.

Weight = Total discharges/Total discharges in 2020 NIS.

The unweighted sample was 6.47 million observations and the weighted sample was around 32.3 million discharges for the year 2020. Patients who were admitted with COVID-19 were retrieved with ICD-10 CM codes, and this group was further divided based on race and sex. The chi-square test was used to compare categorical variables, and linear regression was used for continuous variables. For the primary outcome, univariate logistic regression was used to calculate an unadjusted odds ratio for variables of interest. *p* values of ≤0.2 on univariate logistic regression were used to build a multivariate logistic regression model to adjust for potential confounders. A multivariate linear regression model was used for continuous variables (LOS and Total hospital charge). A two-tailed *p*-value of 0.05 was considered significant.

## 3. Results

Our initial search found 1,659,040 adult inpatient hospitalizations for COVID-19. However, 47,888 records were excluded for missing age and/or gender. We analyzed data from 1,611,152 inpatient hospitalizations for COVID-19. Table 1 shows the socio-demographic characteristics of the sample. By gender identity, 48.0% of records were those of women. By self-identified racial identity, 19.1% were Black, 21.5% were Hispanic, 3.3% were Asian/Pacific Islander, 1.0% were Native American, 50.9% were White, and 4.3% identified as others. Women comprised slightly more than half of Black (52.8%) and Native American (51.8%) patients and less than half of Hispanic (46.2%), Asian/Pacific Islander (47.2%) and White (47.3%) patients. The >70 years age group made up the largest proportion of White female (54.8%), White male (51.6%), and Asian/Pacific Islander female (40.4%) groups, whereas the 50–69 years age group made up the largest proportion of other groups. A median household income of <50,000 USD was reported by approximately half of Native American (57.1% males, 58.0% females) and Black (48.5% male, 49.9% female) patients. Based on insurance, Medicare enrollees made up the largest proportion of all groups. The proportion of uninsured (self-pay) were highest among Black (11.0% males, 9.5% females) and Hispanic (9.6% males, 7.2% females) patients. 

Overall, in-hospital mortality was 13.4%. Mortality was higher in males (15.1%) than females (11.6%). This difference in mortality by gender was maintained across all racial identity groups (Table 2). Native American men had nearly two times the odds of in-hospital mortality (aOR 2.0, CI 1.7–2.4) compared to white men (Appendix A, Figure 1A). Hispanic men (aOR 1.4, CI 1.3–1.45), Native American women (aOR 1.4, CI 1.1–1.7), and Asian/Pacific Islander men (aOR 1.3, CI 1.12–1.4) had higher odds of inpatient death compared with White men. There was no statistically significant difference in mortality among Black men and White men (aOR 1.0, CI 0.9–1.0). White women (aOR 0.7, CI 0.7–0.7), Black women (0.7, CI 0.6–0.7), Hispanic women (aOR 0.9, CI 0.8–0.9), and Asian/Pacific Islander women (aOR 0.8, CI 0.8–0.9) had lower odds of in-hospital mortality compared to White men. 

In total, 15.8% of patients required intubation during hospital stay. The intubation rate was higher in men than women in each of the racial identity groups (Table 2). Compared with White men, Native American men had highest odds of intubation (aOR 1.8, CI 1.5–2.1), followed by Asian/Pacific Islander men (aOR 1.5, CI 1.4–1.6), Hispanic men (aOR 1.5, CI 1.4–1.6), men who identified as others (aOR 1.4, CI 1.3–1.5), and Native American women (aOR1.2, CI 1.0–1.4) (Appendix A, Figure 1B). 

As seen in Table 2, the incidence of renal injury such as AKI, and AKI requiring HD, were generally higher in men. Black men had the highest incidence of AKI (44.2%), followed by Black women (32.3%) and White men (32.2%). In multivariate analysis, Black men had highest odds for AKI (aOR 1.9, CI 1.9–2.0, *p* < 0.001) compared to White men (Appendix A, Figure 1C). Odds of AKI for all non-White men were also higher than those for White men, though the difference did not reach statistical significance for Native American men. Notably, Black women had the highest odds of AKI among all women groups. The incidence of AKI requiring HD was also higher among men than women in each racial identity group. The incidence was highest in Black men (5.1%), followed by Native American men (4.1%), with Hispanic and Asian/Pacific Islander men having similar incidence (3.6%). In contrast with trends seen with AKI, the multivariate analysis showed non-white men with similar but higher odds of developing AKI requiring HD during inpatient hospitalization for COVID-19 (Black men vs. White men aOR 1.9, *p* < 0.001; Hispanic men vs. White men aOR 1.9, *p* < 0.001; Asian/Pacific islander men vs. White men aOR 1.7, *p* < 0.001; Native American men vs. White men 1.9 *p* < 0.001; Others men vs. White men aOR 1.7, *p* < 0.001). White women had lower odds of developing AKI requiring HD than White men (aOR 0.5, *p* < 0.001) (Appendix A, Figure 1D). Non-White women had a similar risk to White men of developing AKI requiring HD, with Black women in particular having the highest odds of this complication among women (Black women vs. White men, aOR 1.1, *p* value < 0.001).

Complications such as cardiac arrest, stroke and shock requiring vasopressor support were also more prevalent in men than women in each racial identity group (Figure 1E–G). Incidence of cardiac arrest was higher in non-White men at approximately 4%, whereas White women had the lowest incidence of cardiac arrest (1.7%) (Appendix A, Figure 1F). Multivariate analysis for risk of cardiac arrest and stroke also showed a similar pattern, with non-White men having higher odds of cardiac arrest (Hispanic men vs. White men (aOR 1.8, CI- 1.7–2.0; Asian/Pacific Islander men vs. White men aOR 1.8, CI 1.5–2.1; Native American men vs. White men aOR 1.7, CI 1.3–2.3; Other men vs. White men aOR 1.7, CI 1.4–1.9) and stroke (Black men vs. White men aOR 1.3, *p* < 0.001; Hispanic men vs. White men (aOR 1.2, *p* < 0.001; Asian/Pacific Islander men vs. White men aOR 1.3, *p =* 0.005; Other men vs. White men aOR 1.7, *p* < 0.001 (Appendix A, Figure 1G). The difference in stroke risk did not reach statistical significance for Native American men. Asian/Pacific islander women had the highest risk of stroke (aOR 1.5, *p=* 0.001) and White women had the lowest risk of stroke (aOR 0.6, CI 0.6–0.7) (Appendix A, Figure 1G). 

The mean length of stay was longest for men who identified as other (9.4 days), followed by Hispanic men (9.4 days), Asian men (9.1 days), and Native American men (9.0 days). Hispanic women represented the highest proportion of patients discharged home (75.6%), whereas White women were discharged to SNF/LTAC/Nursing home at the highest rates (30.1%). 

## 4. Discussion

In this study of over one million hospitalizations for COVID-19 in the United States, we analyzed the effect of intersectionality of gender and racial identity on key health outcomes of mortality, intubations, renal injury, cardiac arrests, stroke, and shock requiring vasopressor support. We found significant difference in outcomes during hospitalization for COVID-19 across different gender and racial identities. Overall, men had higher risk of adverse outcomes than women. Factoring in racial identities, Native American men had the highest risk of mortality and intubations, and Black men had the highest risk of AKI. Non-white men on average had higher risks of shock, cardiac arrest, strokes, and AKI requiring HD than White men, though the difference was not statistically significant for Native American men in analysis of risk of stroke and shock. Mirroring these trends, among women, Native American women had higher risk of mortality and intubations, and Black women had highest risk of AKI and AKI requiring HD. Notably, Asian/Pacific islander women had the highest risk of stroke among all groups. 

Disparities in COVID-19 outcomes across gender and racial identities have been well documented in the literature. Various studies have reported higher mortality from COVID-19 among men [8,9,10]. Our findings are in line with these studies. We further noted that Native American men had the highest mortality risk in our study. While some large-scale studies did not report on mortality rates among Native American subjects [11,12,13], our results are similar to findings from a smaller study in Mississippi, which showed higher mortality among American Indian and Alaska Native patients hospitalized for COVID-19 [14] as well as recent CDC data showing 2.1 × risk of death among AI/AN compared to non-Hispanic White people [15]. While some studies report a higher mortality among Black people [13,16,17], another study by Wong et al. has noted a lower mortality among Black people [18]. Our study found similar risk of mortality among Black men compared to White men. This discrepancy could be a result of different timing of sampling in each study. Time-based trends in COVID-19 infection in different racial identity groups have been noted in a study of US veterans, with Black veterans having a higher proportion of test positivity earlier in the pandemic, whereas the trends changed in summer and fall of 2020 [18]. 

The risk of renal injury was also higher in men than in women. When racial identity groups were considered, Black men had the highest overall risk of renal injury and Black women had highest risk of renal injury among women. This is consistent with existing literature, where male gender and Black racial identity have been associated with AKI in COVID-19 patients [17,19,20]. Several factors could be driving the disparities in renal injury among Black patients—including disparities in baseline health status [21], social determinants of health, historic structural inequities, and genotype differences [22]. Further studies would be needed to elucidate the drivers of this health disparity. 

Our finding of higher risk of adverse outcome among men is consistent with multiple studies that show male gender is associated with poorer outcome in COVID-19 [12,23,24,25]. These differences could be a result of biological differences in susceptibility to viral infection among men due to differences in innate immunity [26], and the effect of androgens on expression ACE2 and TMPRSS2 [27], which are sites of SARS CoV-2 entry into cells. Other factors such as differences in social roles, economic equity, and existing co-morbidities may also contribute to difference in outcomes for COVID-19 among different gender groups. 

There are some key strengths to this study. Consisting of more than 1.6 million hospitalization records, it is one of the largest samples used to examine the intersectional effect of racial and gender identity on outcomes of COVID-19 hospitalization. With the inclusion of >16,500 Native American patient hospitalizations, it also examines the health disparities in this key group, which has been disproportionately affected by COVID-19. 

Our study has several limitations. There is potential sampling bias, as we rely on accurate coding for COVID-19 diagnosis, which was challenging in 2020 with ICD-10 codes developed only in late 2020, raising the possibility of underreporting of cases from early in the pandemic. Gender identity have been reported as a binary; therefore, people with non-binary gender identities could not be included in this study. Racial and ethnic identity has been captured as discrete values and may not represent the true complexity of self-identified race, perceived race and ancestry—each of which may impact health outcomes in different ways. 

## 5. Conclusions

We note that the intersection of gender and racial identities are associated with significant disparities in the outcomes of patients hospitalized for COVID-19 in the United States. BIPOC men have higher risks of adverse outcomes. Further studies are needed to elucidate the various drivers of this disparity and target interventions such as access to care and early vaccination to decrease the health inequities in this population. 

## Figures and Tables

**Figure 1 vaccines-10-02036-f001:**
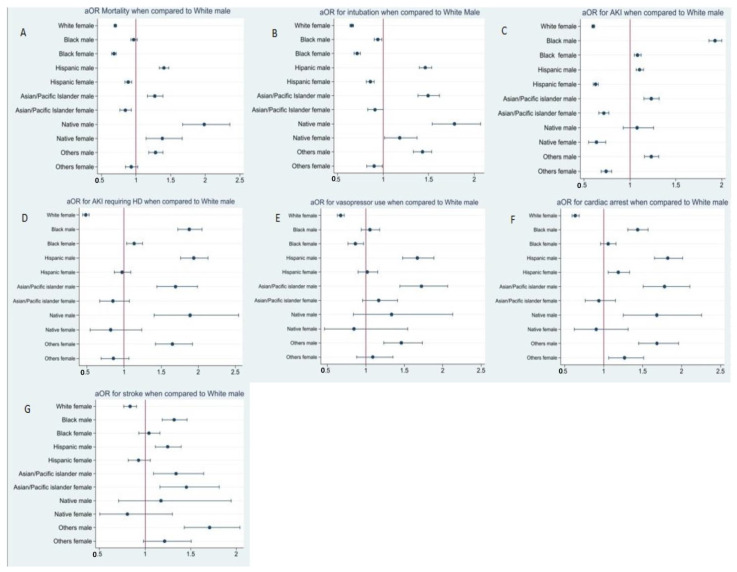
Figure showing adjusted odds risk ratio when compared to white men for (**A**) mortality, (**B**) intubation, (**C**) AKI, (**D**) AKI requiring HD, (**E**) vasopressor, (**F**) cardiac arrest, (**G**) stroke, respectively.

**Table 1 vaccines-10-02036-t001:** Baseline characteristics of hospitalized patients with COVID-19 infection.

	White (820,290)		Black (306,991)		Hispanics (345,935)		Asians (52,326)		Native Americans (16,560)		Others (69,050)		*p* Value
Variables	Male	Female	Male	Female	Male	Female	Male	Female	Male	Female	Male	Female	
**N (%)**	432,293 (52.7%)	387,997 (47.3%)	144,900 (47.2%)	162,091 (52.8%)	186,113 (53.8%)	159,822 (46.2%)	27,628 (52.8%)	24,698 (47.2%)	7982 (48.2%)	8578 (51.8%)	38,806 (56.2%)	30,244 (43.8%)	<0.001
**Mean Age (SD)**	67.94 (15.08)	68.29 (17.2)	60.05 (15.78)	60.03 (18.1)	57.01 (16.2)	54.76 (19.46)	62.05 (16.2)	62.58 (18.5)	56.86 (16.35)	55.95 (17.26)	59.33 (16.32)	58.51 (19.45)	<0.001
**Age Groups**													
≥18–29	8214 (1.9%)	15,132 (3.9%)	6231 (4.3%)	11,995 (7.4%)	9120 (4.9%)	21,097 (13.2%)	884 (3.2%)	1507 (6.1%)	439 (5.5%)	609 (7.1%)	1630 (4.2%)	3024 (10.0%)	<0.001
30–49	43,662 (10.1%)	41,128 (10.6%)	29,124 (20.1%)	31,446 (19.4%)	52,670 (28.3%)	42,513 (26.6%)	5387 (19.5%)	4396 (17.8%)	2235 (28.0%)	2453 (28.6%)	9119 (23.5%)	6623 (21.9%)	<0.001
50–69	157,355 (36.4%)	119,503 (30.8%)	66,944 (46.2%)	64,998 (40.1%)	80,401 (43.2%)	55,139 (34.5%)	11,991 (43.4%)	8817 (35.7%)	3321 (41.6%)	3526 (41.1%)	16,842 (43.4%)	10,737 (35.5%)	<0.001
≥70	223,062 (51.6%)	212,622 (54.8%)	42,601 (29.4%)	53,652 (33.1%)	44,109 (23.7%)	41,234 (25.8%)	9366 (33.9%)	9978 (40.4%)	1988 (24.9%)	1990 (23.2%)	11,215 (28.9%)	9860 (32.6%)	<0.001
**Median Household Income**													
<50,000$	115,855 (26.8%)	107,087 (27.6%)	70,277 (48.5%)	80,883 (49.9%)	69,048 (37.1%)	62,331 (39.0%)	4227 (15.3%)	3976 (16.1%)	4558 (57.1%)	4975 (58.0%)	10,400 (26.8%)	8468 (28.0%)	<0.001
50,000–64,999$	126,230 (29.2%)	114,459 (29.5%)	25,502 (17.6%)	30,959 (19.1%)	50,623 (27.2%)	42,832 (26.8%)	6133 (22.2%)	5878 (23.8%)	1796 (22.5%)	2024 (23.6%)	9702 (25.0%)	7349 (24.3%)	<0.001
65,000–85,999$	104,615 (24.2%)	92,343 (23.8%)	42,166 (29.1%)	45,385 (28.0%)	43,178 (23.2%)	36,120 (22.6%)	7984 (28.9%)	6965 (28.2%)	1030 (12.9%)	1089 (12.7%)	9236 (23.8%)	7168 (23.7%)	<0.001
>86,000$	85,594 (19.8%)	73,719 (19.0%)	6955 (4.8%)	4863 (3.0%)	23,264 (12.5%)	18,539 (11.6%)	9283 (33.6%)	7903 (32.0%)	599 (7.5%)	489 (5.7%)	9469 (24.4%)	7259 (24.0%)	<0.001
**Insurance status**													
Medicare	27,4506 (63.5%)	257,630 (66.4%)	72,450 (50.0%)	85,421 (52.7%)	63,465 (34.1%)	53,381 (33.4%)	11,106 (40.2%)	10,867 (44.0%)	3456 (43.3%)	3689 (43.0%)	14,746 (38.0%)	12,370 (40.9%)	<0.001
Medicaid	28,531 (6.6%)	29,876 (7.7%)	33,617 (23.2%)	36,308 (22.4%)	46,,342 (24.9%)	53,700 (33.6%)	4752 (17.2%)	5038 (20.4%)	2355 (29.5%)	2788 (32.5%)	8266 (21.3%)	8015 (26.5%)	<0.001
Private	119,745 (27.7%)	94,671 (24.4%)	23,039 (15.9%)	24,962 (15.4%)	58,253 (31.3%)	41,234 (25.8%)	10,803 (39.1%)	8101 (32.8%)	1924 (24.1%)	1870 (21.8%)	12,728 (32.8%)	8378 (27.7%)	<0.001
Self-pay	9510 (2.2%)	6208 (1.6%)	15,939 (11.0%)	15,399 (9.5%)	17,867 (9.6%)	11,507 (7.2%)	967 (3.5%)	692 (2.8%)	247 (3.1%)	232 (2.7%)	3066 (7.9%)	1482 (4.9%)	<0.001
**Hospital bedsize**													
Small	110,667 (25.6%)	101,267 (26.1%)	34,051 (23.5%)	38,415 (23.7%)	41,131 (22.1%)	33,882 (21.2%)	5857 (21.2%)	5458 (22.1%)	1652 (20.7%)	2016 (23.5%)	8110 (20.9%)	6109 (20.2%)	<0.001
Medium	123,204 (28.5%)	111,355 (28.7%)	41,007 (28.3%)	45,710 (28.2%)	56,020 (30.1%)	48,746 (30.5%)	7763 (28.1%)	6915 (28.0%)	2187 (27.4%)	2024 (23.6%)	13,116 (33.8%)	10,404 (34.4%)	<0.001
Large	198,422 (45.9%)	175,375 (45.2%)	69,842 (48.2%)	78,128 (48.2%)	88,962 (47.8%)	77,354 (48.4%)	14,007 (50.7%)	12,324 (49.9%)	4143 (51.9%)	4538 (52.9%)	17,579 (45.3%)	13,731 (45.4%)	<0.001
**Hosptal teaching status**													
Rural	59,656 (13.8%)	55,872 (14.4%)	9998 (6.9%)	11,995 (7.4%)	5583 (3.0%)	5114 (3.2%)	470 (1.7%)	445 (1.8%)	1373 (17.2%)	1784 (20.8%)	1281 (3.3%)	1089 (3.6%)	<0.001
Urban non-teaching	87,755 (20.3%)	79,539 (20.5%)	20,431 (14.1%)	23,179 (14.3%)	37,409 (20.1%)	31,165 (19.5%)	5028 (18.2%)	4470 (18.1%)	1133 (14.2%)	1184 (13.8%)	6403 (16.5%)	4869 (16.1%)	<0.001
Urban teaching	284,881 (65.9%)	252,586 (65.1%)	114,471 (79.0%)	126,917 (78.3%)	143,121 (76.9%)	123,542 (77.3%)	22,158 (80.2%)	19,783 (80.1%)	5476 (68.6%)	5601 (65.3%)	31,161 (80.3%)	24,286 (80.3%)	<0.001
**Comorbidities**													
CAD	123,636 (28.6%)	67,123 (17.3%)	24,343 (16.8%)	21,396 (13.2%)	23,264 (12.5%)	12,946 (8.1%)	4559 (16.5%)	2593 (10.5%)	1317 (16.5%)	695 (8.1%)	6054 (15.6%)	2813 (9.3%)	<0.001
CHF	92,511 (21.4%)	76,823 (19.8%)	30,284 (20.9%)	31,446 (19.4%)	21,031 (11.3%)	15,023 (9.4%)	3675 (13.3%)	2890 (11.7%)	1317 (16.5%)	1089 (12.7%)	4851 (12.5%)	3539 (11.7%)	<0.001
HTN uncomplicated	170,323 (39.4%)	153,647 (39.6%)	55,931 (38.6%)	66,457 (41.0%)	64,953 (34.9%)	54,180 (33.9%)	11,079 (40.1%)	10,052 (40.7%)	2802 (35.1%)	2736 (31.9%)	14,009 (36.1%)	10,979 (36.3%)	<0.001
HTN complicated	131,849 (30.5%)	105,923 (27.3%)	55,062 (38.0%)	51,707 (31.9%)	36,664 (19.7%)	26,530 (16.6%)	4780 (17.3%)	4470 (18.1%)	1996 (25.0%)	1819 (21.2%)	8110 (20.9%)	5474 (18.1%)	<0.001
DM uncomplicated	57,927 (13.4%)	48,888 (12.6%)	22,749 (15.7%)	27,880 (17.2%)	29,964 (16.1%)	26,690 (16.7%)	8316 (30.1%)	6644 (26.9%)	1301 (16.3%)	1570 (18.3%)	6325 (16.3%)	4839 (16.0%)	<0.001
DM complicated	111,099 (25.7%)	82,255 (21.2%)	48,252 (33.3%)	49,924 (30.8%)	54,717 (29.4%)	41,554 (26.0%)	6824 (24.7%)	5359 (21.7%)	2786 (34.9%)	2959 (34.5%)	10,516 (27.1%)	6835 (22.6%)	<0.001
Renal failure	101,157 (23.4%)	74,883 (19.3%)	47,237 (32.6%)	39,712 (24.5%)	31,081 (16.7%)	21,097 (13.2%)	5968 (21.6%)	4446 (18.0%)	1572 (19.7%)	1475 (17.2%)	6830 (17.6%)	4264 (14.1%)	<0.001
Chronic pulmonary disease	105,479 (24.4%)	112,519 (29.0%)	27,676 (19.1%)	41,009 (25.3%)	20,659 (11.1%)	25,572 (16.0%)	4365 (15.8%)	4001 (16.2%)	1261 (15.8%)	2024 (23.6%)	5239 (13.5%)	5293 (17.5%)	<0.001
Obesity	98,131 (22.7%)	99,327 (25.6%)	35,501 (24.5%)	58,839 (36.3%)	47,459 (25.5%)	47,787 (29.9%)	3813 (13.8%)	3606 (14.6%)	2035 (25.5%)	2762 (32.2%)	7645 (19.7%)	7138 (23.6%)	<0.001
Smoking	153,464 (35.5%)	100,103 (25.8%)	42,456 (29.3%)	32,580 (20.1%)	40,945 (22.0%)	17,421 (10.9%)	6990 (25.3%)	2149 (8.7%)	2315 (29.0%)	2162 (25.2%)	8576 (22.1%)	3508 (11.6%)	<0.001

**Table 2 vaccines-10-02036-t002:** In-hospital outcomes of patients with COVID-19 infection.

	White (820,290)		Black (306,991)		Hispanics (345,935)		Asians (52,326)		Native (16,560)		Others (69,050)		*p* Value
	Male	Female	Male	Female	Male	Female	Male	Female	Male	Female	Male	Female	
**Disposition**													
Home/Routine	242,516 (56.1%)	199,430 (51.4%)	87,809 (60.6%)	99,524 (61.4%)	137,165 (73.7%)	120,825 (75.6%)	18,842 (68.2%)	16,449 (66.6%)	5532 (69.3%)	6313 (73.6%)	26,388 (68.0%)	20,717 (68.5%)	<0.001
SNF/LTAC/Nursing home	111,532 (25.8%)	116,787 (30.1%)	33,907 (23.4%)	32,094 (19.8%)	23,450 (12.6%)	17,421 (10.9%)	4448 (16.1%)	4174 (16.9%)	1477 (18.5%)	1227 (14.3%)	6558 (16.9%)	4960 (16.4%)	<0.001
Home health	71,761 (16.6%)	68,675 (17.7%)	19,562 (13.5%)	28,366 (17.5%)	21,961 (11.8%)	19,658 (12.3%)	4006 (14.5%)	3927 (15.9%)	790 (9.9%)	884 (10.3%)	5006 (12.9%)	4264 (14.1%)	<0.001
AMA	6052 (1.4%)	3104 (0.8%)	3623 (2.5%)	2107 (1.3%)	3350 (1.8%)	1758 (1.1%)	332 (1.2%)	123 (0.5%)	184 (2.3%)	154 (1.8%)	776 (2.0%)	302 (1.0%)	0.17
**AKI**	138,766 (32.1%)	92,731 (23.9%)	64,046 (44.2%)	52,355 (32.3%)	48,762 (26.2%)	27,649 (17.3%)	8758 (31.7%)	5557 (22.5%)	2219 (27.8%)	1673 (19.5%)	11,758 (30.3%)	6563 (21.7%)	<0.001
**AKI with HD**	9078 (2.1%)	4268 (1.1%)	7390 (5.1%)	5025 (3.1%)	6700 (3.6%)	2877 (1.8%)	995 (3.6%)	395 (1.6%)	327 (4.1%)	163 (1.9%)	1358 (3.5%)	514 (1.7%)	<0.001
**Cardiac arrest**	10,807 (2.5%)	6596 (1.7%)	5941 (4.1%)	4863 (3.0%)	7631 (4.1%)	4155 (2.6%)	1077 (3.9%)	519 (2.1%)	295 (3.7%)	172 (2.0%)	1513 (3.9%)	877 (2.9%)	<0.001
**Intubation**	72,193 (16.7%)	46,948 (12.1%)	27,096 (18.7%)	23,827 (14.7%)	37,781 (20.3%)	20,937 (13.1%)	5940 (21.5%)	3433 (13.9%)	2099 (26.3%)	1681 (19.6%)	8110 (20.9%)	4355 (14.4%)	<0.001
**Vasopressor use**	10,807 (2.5%)	6596 (1.7%)	4782 (3.3%)	4214 (2.6%)	7072 (3.8%)	3676 (2.3%)	1326 (4.8%)	766 (3.1%)	216 (2.7%)	146 (1.7%)	1475 (3.8%)	877 (2.9%)	<0.001
**Stroke**	6917 (1.6%)	5044 (1.3%)	3043 (2.1%)	2593 (1.6%)	2978 (1.6%)	1918 (1.2%)	525 (1.9%)	469 (1.9%)	112 (1.4%)	77 (0.9%)	893 (2.3%)	484 (1.6%)	<0.001
**Died**	66,573 (15.4%)	47,724 (12.3%)	19,996 (13.8%)	17,506 (10.8%)	27,359 (14.7%)	16,302 (10.2%)	4476 (16.2%)	2964 (12.0%)	1509 (18.9%)	1209 (14.1%)	6054 (15.6%)	3750 (12.4%)	<0.001
**Mean LOS**	7.9	7.2	9.0	8.0	9.4	7.3	9.1	7.6	9.0	8.2	9.4	7.6	<0.001
**Mean TOTCHG**	84,639.4	69,127	98,138.8	82,105.2	136,746	99,750	12,8947.9	98,974.5	106,172	88,653	132,199.6	101,390.5	<0.001

## Data Availability

Not applicable.

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
