# Peer review of "Gender and Race-Based Health Disparities in COVID-19 Outcomes among Hospitalized Patients in the United States: A Retrospective Analysis of a National Sample"

_vaccines, 2022, doi:10.3390/vaccines10122036_

Round 1

Reviewer 1 Report

Thank you for the opportunity to review this engaging manuscript. The use of the AHRQ NIS data allows for a comprehensive overview of the effects of the pandemic, and I anticipate that this study will attractive significant citation based on its relevant to the pandemic and overall human health.

I have no major concerns regarding the content of the manuscript.  The data is clearly presented, and the overall writing and message are clear. Most of my recommendations are more minor points relating to formatting, language and clarity.  Specific comments are enumerated below:

Major comment: 

The authors observe that Black men had mortality similar to white men in the NIS database.  This is a highly unexpected finding given the widely publicized higher rates of mortality reported in the media and in many other scientific studies.  The authors write, "There was no statistically significant difference in mortality among Black men and White men." (line 100).  Please provide odds ratio and confidence interval for this observation.  Also, it would be important for readers to understand if Black men were dying younger, despite not having an overall higher mortality rate.

Minor comments:

1. Although a minor point, I suggest reducing digits after the decimal place for sociodemographic data.  I recommend one digit after the decimal point (e.g., 32.2%, not 32.165%) and similar approach for adjusted odds ratios and for confidence intervals (CI 1.5-2.1 instead of 1.53 to 2.06) to avoid distracting readers and overstating the precision of estimates.

2. The presentation of dollar amount on line 90, "<49,999$ is not conventional.  I would suggest "<$50,000 for clarity.

3. The author refer to weighted versus unweighted counts at the beginning of their report.  Please briefly explain the methodology used to arrive at weighted and unweighted figures.

4. There are some minor grammar/ English syntax issues that appear in the paper.  For example, on line 100 please correct the noun/verb agreement error -- "Asian/Pacific Islander men (aOR 1.27, CI 1.16-1.38) has higher odds" should read "...have higher odds"

5. Please correct formatting for capitalization:

line 142 -- should use "aOR" (not "Aor") for adjusted odds ratio in the following excerpt  "Asian/Pacific Islander men vs White men aOR 1.34, p 141 0.005; Other men vs White men Aor 1.70, p <0.001 (Supplemental table 6, figure 1F)."

6. line 142: What is "fAKI"? Perhaps there were some words inadvertently deleted in this sentence (?)

6. line 184 - There is unnecessary capitalization of the word "is" at line 185 "This Is consistent with existing 184literature where male gender and Black racial identity have been associated with AKI in COVID-19 patients17,19,20."

7. Causality should not be implied in studies that are measuring associations

   line 15:  The authors state, "Racial and gender identity are determinants of prevalent healthcare disparities."  Racial and gender identity are linked, or associated, with disparities (and Social Determinants of Health contribute to health inequity); but it is preferable to note the existence of an association rather than to state that Racial and gender identity are determinants of prevalent healthcare disparities.

   Similarly, in line 211, the authors state, "We note that the intersection of gender and racial identities have significant impact on outcomes of patients hospitalized for COVID-19 in the United States." There is an association present, but we cannot know whether race or gender directly cause (or impact) the outcomes, so the word association is preferred.

Author Response

Dear Respectful Reviewer: 

Thank you for your valuable feedback and comments. Please see below our response to every query/comment. 

Thank you for the opportunity to review this engaging manuscript. The use of the AHRQ NIS data allows for a comprehensive overview of the effects of the pandemic, and I anticipate that this study will attractive significant citation based on its relevant to the pandemic and overall human health.

I have no major concerns regarding the content of the manuscript.  The data is clearly presented, and the overall writing and message are clear. Most of my recommendations are more minor points relating to formatting, language and clarity.  Specific comments are enumerated below:

Major comment: 

The authors observe that Black men had mortality similar to white men in the NIS database.  This is a highly unexpected finding given the widely publicized higher rates of mortality reported in the media and in many other scientific studies.  The authors write, "There was no statistically significant difference in mortality among Black men and White men." (line 100).  Please provide odds ratio and confidence interval for this observation.  Also, it would be important for readers to understand if Black men were dying younger, despite not having an overall higher mortality rate.

Response - We have provided the odds ratio and confidence interval for this observation.

This finding was a surprise to us as well, specially given ( as the reviewer has noted ) prior publicisized findings of higher COVID-19 mortality in Black patients,. But our review of literature on the subject, as we note in our discussion, found a mixed picture with some studies reporting higher mortality among Black patients with COVID-19, however other studies have not found this to be the case. We hypothesized that the difference in findings could be due to differential timing of the studies. Time based trends in COVID-19 infection among different racial identity groups has been noted in at least one study of US veterans that we cite in our discussion. It should also be noted that our study examined only in-hospital deaths of patients with COVID-19 and does not include all deaths due to COVID-19 which may include deaths in the community, at nursing facilities or other institutions, correctional facilities, etc. However, this finding certainly deserves a closer examination.

We did not conduct an age specific mortality rate analysis therefore cannot answer the reviewers query of whether Black men were dying younger, but this could be an area of future investigation.

Minor comments:

1. Although a minor point, I suggest reducing digits after the decimal place for sociodemographic data.  I recommend one digit after the decimal point (e.g., 32.2%, not 32.165%) and similar approach for adjusted odds ratios and for confidence intervals (CI 1.5-2.1 instead of 1.53 to 2.06) to avoid distracting readers and overstating the precision of estimates.

Response – We accept the reviewers suggestion and have made the changes to the manuscript so that all data is now presented to rounded to one decimal place.

2. The presentation of dollar amount on line 90, "<49,999$ is not conventional.  I would suggest "<$50,000 for clarity.

Response -We accept the reviewers suggestion and have made the edits to the manuscript text

3. The author refer to weighted versus unweighted counts at the beginning of their report.  Please briefly explain the methodology used to arrive at weighted and unweighted figures.

Response –  We have included a description in the methods section which reads as follows -

NIS 2020 data is sampled from 49 statewide organizations covering 98% of US population and 97% of discharges from US community hospitals. Sampled data includes approximates to 20% or ~7 million discharges collected for that year. To obtain national estimates and extrapolate results for particular disease, weights were provided and they are calculated as using below formula. Ideally each disease discharge weight should be 5, but the weight significantly varies due to random sampling and hence we use weights provided using survey analysis to provide national estimates.

Weight = Total discharges/ Total discharges in 2020 NIS.

  1. There are some minor grammar/ English syntax issues that appear in the paper.  For example, on line 100 please correct the noun/verb agreement error -- "Asian/Pacific Islander men (aOR 1.27, CI 1.16-1.38) has higher odds" should read "...have higher odds"

Response – We have reviewed the manuscript for grammar and syntax and have made corrections as suggested by the reviewer.

  1. Please correct formatting for capitalization:

line 142 -- should use "aOR" (not "Aor") for adjusted odds ratio in the following excerpt  "Asian/Pacific Islander men vs White men aOR 1.34, p 141 0.005; Other men vs White men Aor 1.70, p <0.001 (Supplemental table 6, figure 1F)."

Response – We thank the reviewer for catching this typographical error. This has been corrected.  

  1. line 142: What is "fAKI"? Perhaps there were some words inadvertently deleted in this sentence (?)

Response – We thank the reviewer for catching this typographical error. “fAKI” has been deleted. It was overlooked in our previous edit of the manuscript.

  1. line 184 - There is unnecessary capitalization of the word "is" at line 185 "This Is consistent with existing 184literature where male gender and Black racial identity have been associated with AKI in COVID-19 patients17,19,20."

Response – The capitalization has been removed.

  1. Causality should not be implied in studies that are measuring associations

   line 15:  The authors state, "Racial and gender identity are determinants of prevalent healthcare disparities."  Racial and gender identity are linked, or associated, with disparities (and Social Determinants of Health contribute to health inequity); but it is preferable to note the existence of an association rather than to state that Racial and gender identity are determinants of prevalent healthcare disparities.

   Similarly, in line 211, the authors state, "We note that the intersection of gender and racial identities have significant impact on outcomes of patients hospitalized for COVID-19 in the United States." There is an association present, but we cannot know whether race or gender directly cause (or impact) the outcomes, so the word association is preferred.

Response – We accept the reviewers premise that association does not imply causation and have edited both sentences to reflect the same.

Reviewer 2 Report

the paper describes the relationship between some variables and the outcomes of hospitalized for Covid in the USA. The work is well structured and recruits a very significant number of cases which gives extreme significance to the results. It is good that the authors clarify why they have chosen renal failure, which is not a frequent complication in Covid, among the elements to be considered from a clinical point of view.

Author Response

Dear Respectful Reviewer,

Thank you for your feedback and comments. 

the paper describes the relationship between some variables and the outcomes of hospitalized for Covid in the USA. The work is well structured and recruits a very significant number of cases which gives extreme significance to the results. It is good that the authors clarify why they have chosen renal failure, which is not a frequent complication in Covid, among the elements to be considered from a clinical point of view.

Response – We thank the reviewer for their careful consideration of our work and the positive comments.

Reviewer 3 Report

At present, many similar papers have been published, and the subjects of this study were included from January 1, 2020 to December 31, 2020. The  Delta or Omicron COVID-19 variants in 2021-2022 were not included in the analysis of gender and race based health disparities; Therefore, the innovation of this study is not enough, and there are no new findings compared with the similar studies.. For example:

(1)Racial and Ethnic Disparities in Excess Deaths During the COVID-19 Pandemic, March to December 2020. Ann Intern Med. 2021 Dec;174(12):1693-1699.  In this paper,Excess deaths and excess deaths per 100 000 persons from March to December 2020 were estimated by race/ethnicity, sex, age group, and cause of death, using provisional death certificate data from the Centers for Disease Control and Prevention (CDC) and U.S. Census Bureau population estimates.

(2)Sex-, Race- and Ethnicity-Based Differences in Thromboembolic Events Among Adults Hospitalized With COVID-19. J Am Heart Assoc. 2021 Dec 7;10(23):e022829.

(3)The influence of sex, gender, age, and ethnicity on psychosocial factors and substance use throughout phases of the COVID-19 pandemic. PLoS One. 2021 Nov 22;16(11):e0259676.

(4)Gender and Ethnic Disparities of Acute Kidney Injury in COVID-19 Infected Patients: A Literature Review. Front Cell Infect Microbiol. 2022 Jan 13;11:778636.

Author Response

Dear Respectful Reviewer:

Thank you for your feedback and comments. 

At present, many similar papers have been published, and the subjects of this study were included from January 1, 2020 to December 31, 2020. The  Delta or Omicron COVID-19 variants in 2021-2022 were not included in the analysis of gender and race based health disparities; Therefore, the innovation of this study is not enough, and there are no new findings compared with the similar studies.. For example:

(1)Racial and Ethnic Disparities in Excess Deaths During the COVID-19 Pandemic, March to December 2020. Ann Intern Med. 2021 Dec;174(12):1693-1699.  In this paper,Excess deaths and excess deaths per 100 000 persons from March to December 2020 were estimated by race/ethnicity, sex, age group, and cause of death, using provisional death certificate data from the Centers for Disease Control and Prevention (CDC) and U.S. Census Bureau population estimates.

(2)Sex-, Race- and Ethnicity-Based Differences in Thromboembolic Events Among Adults Hospitalized With COVID-19. J Am Heart Assoc. 2021 Dec 7;10(23):e022829.

(3)The influence of sex, gender, age, and ethnicity on psychosocial factors and substance use throughout phases of the COVID-19 pandemic. PLoS One. 2021 Nov 22;16(11):e0259676.

(4)Gender and Ethnic Disparities of Acute Kidney Injury in COVID-19 Infected Patients: A Literature Review. Front Cell Infect Microbiol. 2022 Jan 13;11:778636.

Response – We thank the reviewer for their careful consideration of our manuscript.

The time period of our study corresponds to available NIS data, which does not include the time period of Delta and Omicron variants for which NIS data is not currently available.

We believe our manuscript has several strengths over each of the studies mentioned.

  • DOI: 7326/M21-2134 – The study by Shiels et al. is an excellent study examining the burden of COVID-19 pandemic by excess deaths. It, however, does not specifically look at outcomes from hospitalization for COVID-19, which is the focus of our study.
  • DOI: 1161/JAHA.121.022829 – Llyas et al have examined thrombo-embolic events in hospitalized COVID-19 patients, but have a much smaller sample size than our study. Specifically, their sample had a much lower Native American population than in our study.

We do note the importance of their findings and have already cited them in our discussion.

  • DOI: 1371/journal.pone.0259676 This study by Brotto et al is different from our study due to the following reasons –
  1. The study population is Canadian
  2. The study population is a population registry and not a population of patients hospitalized for COVID-19
  3. The outcomes studied are psychological effects of the pandemic, and not clinical outcomes of COVID-19 infection and hospitalization for the same.

  • DOI: 10.3389/fcimb.2021.778636 – The study by He et al is excellent but it is a literature review and does not provide primary data. It also examines a single complication of AKI whereas we have analyzed several key outcomes of interest. Lastly, it does not focus on the US population as we have done.

We hope that we have clarified why our study, building on prior literature, provides new insights into this area and, by its inclusion of a large sample of Native American patients, provides key data into the disparities affecting this population.

Round 2

Reviewer 3 Report

The author has answered why the author wants to carry out this study based on the previous similar studies. The author has explained and revised the paper in detail. I think the revised paper is acceptable, but there are several minor comments: (1) It is suggested to add all the percentage of cases in Table 1 and Table 2; (2) The sample size N of all races is missing in table 2, (3) Some P-values are missing in Supplemental table 4

Author Response

Dear reviewer, Thanks for your valuable comment, We have modified the manuscript as requested. 

(1) It is suggested to add all the percentages of cases in Table 1 and Table 2

- added, we also mentioned n and % in the bracket. 

; (2) The sample size N of all races is missing in table 2,

Added 

(3) Some P-values are missing in Supplemental table 4

Added 

Regards 

RS